# Using deep-learning in fetal ultrasound analysis for diagnosis of cystic hygroma in the first trimester

Mark C. Walker[1,2,3,4,5,6]*, Inbal Willner[1,5], Olivier X. Miguel[2], Malia S. Q. Murphy[2], Darine El-Chaâr[1,2,4,5], Felipe Moretti[1,5], Alysha L. J. Dingwall Harvey[2], Ruth Rennicks White[2,5], Katherine A. Muldoon[1,2], André M. Carrington[7,8], Steven Hawken[2,4], Richard I. Aviv[8,9,10]

1 Department of Obstetrics and Gynecology, University of Ottawa, Ottawa, Canada, 2 Clinical Epidemiology Program, Ottawa Hospital Research Institute, Ottawa, Canada, 3 International and Global Health Office, University of Ottawa, Ottawa, Canada, 4 School of Epidemiology and Public Health, University of Ottawa, Ottawa, Canada, 5 Department of Obstetrics, Gynecology & Newborn Care, The Ottawa Hospital, Ottawa, Canada, 6 BORN Ontario, Children's Hospital of Eastern Ontario Research Institute, Ottawa, Canada, 7 Department of Systems Design Engineering, University of Waterloo, Waterloo, Canada, 8 Department of Radiology and Medical Imaging, University of Ottawa, Ottawa, Canada, 9 Department of Radiology and Medical Imaging, The Ottawa Hospital, Ottawa, Canada, 10 Neuroscience Program, Ottawa Hospital Research Institute, Ottawa, Canada

* mwalker@toh.ca

## Abstract

### Objective

To develop and internally validate a deep-learning algorithm from fetal ultrasound images for the diagnosis of cystic hygromas in the first trimester.

### Methods

All first trimester ultrasound scans with a diagnosis of a cystic hygroma between 11 and 14 weeks gestation at our tertiary care centre in Ontario, Canada were studied. Ultrasound scans with normal nuchal translucency were used as controls. The dataset was partitioned with 75% of images used for model training and 25% used for model validation. Images were analyzed using a DenseNet model and the accuracy of the trained model to correctly identify cases of cystic hygroma was assessed by calculating sensitivity, specificity, and the area under the receiver-operating characteristic (ROC) curve. Gradient class activation heat maps (Grad-CAM) were generated to assess model interpretability.

### Results

The dataset included 289 sagittal fetal ultrasound images;129 cystic hygroma cases and 160 normal NT controls. Overall model accuracy was 93% (95% CI: 88–98%), sensitivity 92% (95% CI: 79–100%), specificity 94% (95% CI: 91–96%), and the area under the ROC curve 0.94 (95% CI: 0.89–1.0). Grad-CAM heat maps demonstrated that the model predictions were driven primarily by the fetal posterior cervical area.

**Data Availability Statement:** The ultrasound images used in this study are not currently available to upload to PLOS ONE. All images belong to The Ottawa Hospital. A formal request can be

made to the IRB, who would then consult a privacy officer at The Ottawa Hospital. Requests can be sent to the corresponding author (Dr. Mark Walker) or the Ottawa Health Sciences Network Research Ethics Board (rebadministration@ohri. ca)/(http://www.ohri.ca/ohsn-reb/contacts.htm).

**Funding:** This study was supported by a Canadian Institutes of Health Research Foundation Grant (FDN 148438)(MCW). The funding agency played no role in the study design, data collection and analysis, decision to publish, or preparation of the manuscript.

**Competing interests:** The authors have declared that no competing interests exist.

## Conclusions

Our findings demonstrate that deep-learning algorithms can achieve high accuracy in diagnostic interpretation of cystic hygroma in the first trimester, validated against expert clinical assessment.

## Introduction

Artificial intelligence (AI) and machine learning models are increasingly being applied in clinical diagnostics, including medical imaging [1]. Deep-learning is a class of machine learning models inspired by artificial neural networks which can process large amounts of data to identify important features that are predictive of outcomes of interest. Importantly, model performance can be continuously and incrementally improved as data accrues. Deep-learning models perform well in pattern recognition, making them particularly useful in the interpretation of medical images [2]. A systematic review and meta-analysis of studies reporting on the diagnostic performance of deep-learning models in the identification of disease features from medical images found that they performed equivalently to trained clinicians, however, few studies make direct comparisons between deep-learning models and health care professionals [2].

AI has many potential uses in ultrasonography, including automating standardized plane detection, and extracting quantitative information about anatomical structure and function. Inter-operator heterogeneity in ultrasound image acquisition poses unique challenges for the use of AI in this space, and are further impeded by a lack of standardized imaging planes [3]. Still, AI has been successfully tested in a range of organ systems, and there is growing interest for its application in obstetric ultrasonography for fetal structure identification, automated measurement of fetal growth parameters, and for the diagnosis of congenital anomalies [4]. Uptake of AI applications into clinical practice is beginning, however, ongoing investigation is warranted to demonstrate the feasibility, validity and reliability of such tools.

Cystic hygroma is a congenital lymphangioma documented in approximately 1 in 800 pregnancies and 1 in 8 000 live births [5]. It is commonly associated with chromosomal abnormalities including Trisomy 21, Turner Syndrome, and anatomical malformations [6–8]. Cystic hygroma is diagnosed based on the assessment of nuchal translucency (NT) thickness between the fetal skin and the subcutaneous soft tissue at the neck and cervical spine [4]. Diagnosis is straightforward and possible through routine first- or second-trimester ultrasonography. Affected pregnancies require extensive antenatal and postpartum management which may include early cytogenetic testing for suspected aneuploidy, comprehensive assessment of fetal anatomy, and postpartum surgical intervention [8, 9]. Some studies have investigated automated and semi-automated systems for measuring NT thickness, however, no studies to date have investigated AI and deep-learning models to assess diagnoses associated with NT thickness, including cystic hygroma [10, 11].

Given the unique challenges in applying AI in fetal ultrasonography, cystic hygroma is an ideal condition to investigate the feasibility of using deep-learning models in the interpretation of ultrasound images. We sought to develop a deep-learning model that could analyze fetal ultrasound images and correctly identify cases of cystic hygroma compared to normal controls.

## Methods

### Study setting and design

This was a retrospective study conducted at The Ottawa Hospital, a multi-site tertiary-care facility in Ottawa, Canada, with a catchment area of 1.3 million people. First trimester

A                                    B

**Fig 1.** Fetal ultrasound images of normal (A) and cystic hygroma (B) scans.

ultrasound images taken between March 2014 and March 2021 were retrieved from the institutional Picture Archiving and Communication System (PACS) and saved in Digital Imaging and Communications in Medicine (DICOM) format. Eligible images were those that included a mid-sagittal view of the fetus taken between 11 and 14 weeks gestation. Cases were identified if there was a final clinical diagnosis of a cystic hygroma in the ultrasound report (Fig 1A). A set of normal ultrasound images from NT screens were used for controls and were retrieved between March 2021 and June 2021 (Fig 1B). Cases and normal images were reviewed and verified by a clinical expert (IW). Patients were not contacted and patient consent was not required to access the images. Data were de-identified and fully anonymized for the model training. This study was reviewed and approved by the Ottawa Health Sciences Network Research Ethics Board (OHSN REB #20210079).

A 4-fold cross-validation (4CV) design was used, whereby the same deep-learning architecture was tested and trained four different times using randomly partitioned versions (folds) of the image dataset. For each fold, 75% of the dataset was used for model training, and 25% was used for model validation. The 4CV design was chosen instead of the more commonly used 10-fold cross-validation (10CV) design to optimize the performance of the deep-learning models within our small dataset. With 4CV, each prediction error affects the accuracy of the model by 1.4%, and the sensitivity by 3.1%. Had a 10CV approach been used, each prediction error would have affected model accuracy by 3.4% and the sensitivity by 7.8%.

## Data preparation

DICOM images included coloured annotations such as calipers, text, icons, profile traces (Fig 2A) and patient personal health information (PHI), which were removed prior to analysis. PHI was handled by cropping the images to remove the identifying information contained in the image borders. Coloured annotations were removed by first converting image data from the Red Green Blue (RGB) colour space to the Hue Saturation Value (HSV) colour space. Image pixels belonging to the grey ultrasound image were empirically identified and ranged from 0–27, 0–150 and 0–255 for the H, S and V values, respectively (Fig 2B).

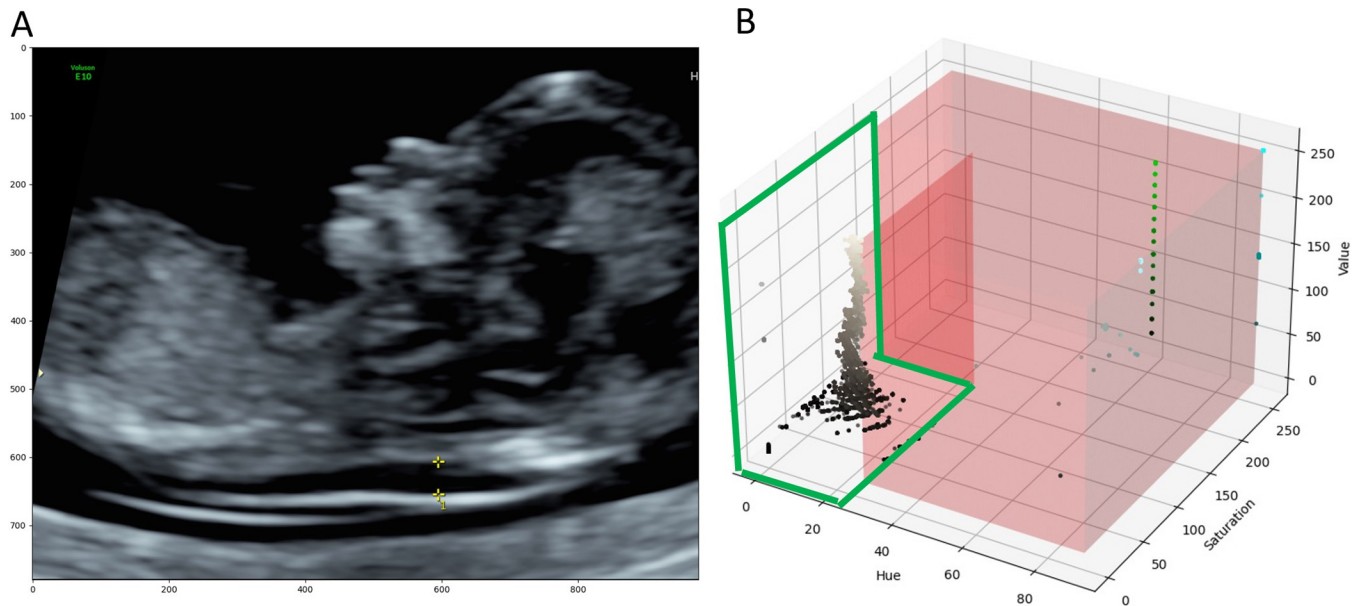

**Fig 2. Identifying image annotations on a normal NT scan.** (A) Image annotations included calipers, text, icons, and profile traces, all of which were removed prior to model training. (B) 3D Scatter Plot of HSV image data. Each point represents one image pixel and its associated HSV values. The red region highlights the range of values which do not belong to the grayscale ultrasound image. The area encircled in green shows pixel values that belong to the grayscale ultrasound image. Grayscale images had H, S and V values ranging from 0–27, 0–150 and 0–255, respectively.

Pixels outside of these ranges were determined to be part of the coloured annotations. Third, a binary mask of the image was created where annotation pixels were labelled '1' and ultrasound image pixels were labelled '0'. The binary mask was then dilated with a 5x5 kernel to include contours surrounding the annotations. Last, the Navier-Stokes image infill method [12] was used to artificially reconstruct the ultrasound image without annotations (Fig 3).

After DICOM images were cleaned, they were converted to grayscale (1 channel image). Intensities were standardized to a mean of zero and a standard deviation of one for better stability during the neural network training [13]. Finally, the images were resized to 256 x 256 pixels.

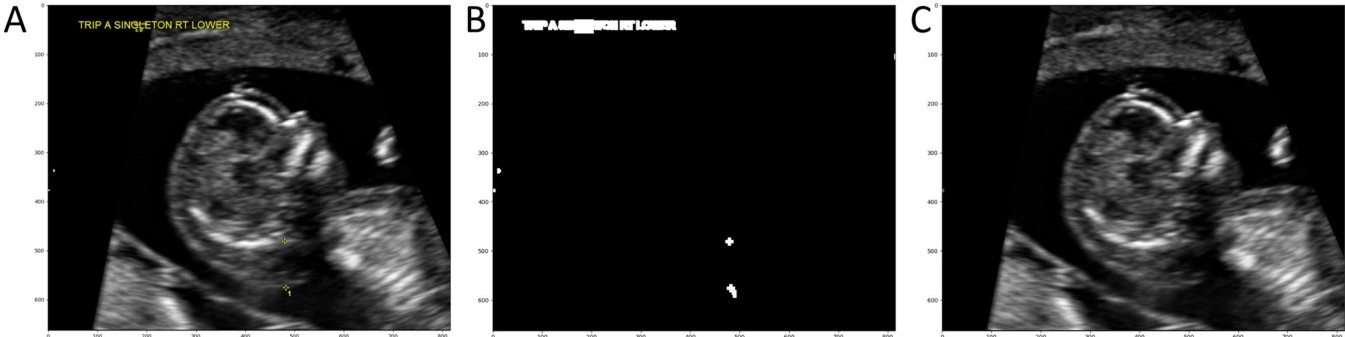

**Fig 3. Removal of image annotations on a scan with cystic hygroma diagnosis.** (A) Ultrasound image before annotations were removed. Yellow calipers (bottom middle) are visible, along with text annotations (top left). (B) The binary mask of the image which was generated to define the region of the image that need to be infilled (white pixels). (C) Result of the Navier-Stokes image infill method; all image annotations have been removed.

## Neural network model

A DenseNet convolutional neural network (CNN) model architecture[14] was used to classify images as "normal" or those with "cystic hygroma". Specifically, we used the DenseNet169 PyTorch model [15], with the input layer modified to have a shape of 256 x 256 x 1 pixels and the output layer modified to produce two possible outcome features: 1) normal and 2) cystic hygroma. DenseNet169 was chosen because of its advantages over the ResNet architecture as shown in a publication by Gao et al. (2017) demonstrating that DenseNet achieves high performance while requiring less computation [12]. On some image classification benchmarks DenseNet demonstrated better performance and efficiency than the ResNet model.

## Model training

Model training was performed from scratch with random weights initialization for 1000 epochs (i.e., step-in optimization) using the cross-entropy loss function [16], the Adam optimizer [17, 18] (epsilon = 1e-8, beta1 = 0.9, beta2 = 0.999) and a batch size of 64. The available pretrained DenseNet models were trained on ImageNet. ImageNet and fetal ultrasound images are significantly different, therefore CNN models for fetal US applications should not be pretrained on ImageNet. We trained the architecture using our dataset of ultrasound images from The Ottawa Hospital with random weights initialization. The learning rate was set to 1e-2 and reduced by a factor of 0.72 at every 100 epochs (gamma = 0.72, step size = 100) using a learning rate step scheduler [19].

Data augmentation was used during model training to increase the generalizability of the CNN model. Augmentations included random horizontal flip of the ultrasound images (50% probability), random rotation in the range of [–15, 15] degrees, random translation (x ±10% and y±30% of image size) and random shearing in the range of [-0.2, 0.2] degrees. These augmentations were performed dynamically at batch loading. Due to the randomness of these operations, a single training image underwent different augmentations at each epoch.

To address imbalance in the number of normal NT and cystic hygroma images in the training dataset, cystic hygroma images were randomly up sampled, with replacement, to match the number of normal NT images.

## Model validation

For each epoch, the validation data were used to assess the total number of true and false positives, and true and false negatives and used to calculate accuracy, sensitivity, specificity and the area under the receiver-operating characteristic curve (AUC). For each fold, the performance metrics for the epoch with the highest level of accuracy were reported. The mean, standard deviation and 95% confidence intervals (CI) of the performance metrics across all folds were then computed.

## Explainability

The Gradient-weighted Class Activation Mapping (Grad-CAM) method was used to improve the interpretability of our trained DenseNet models, and visually contextualize important features in the image data that were used for model predictions [20]. Grad-CAM is a widely used technique for the visual explanation of deep-learning algorithms [21, 22]. With this approach, heat maps were generated from 8x8 feature maps to highlight regions of each image that were the most important for model prediction (Fig 4).

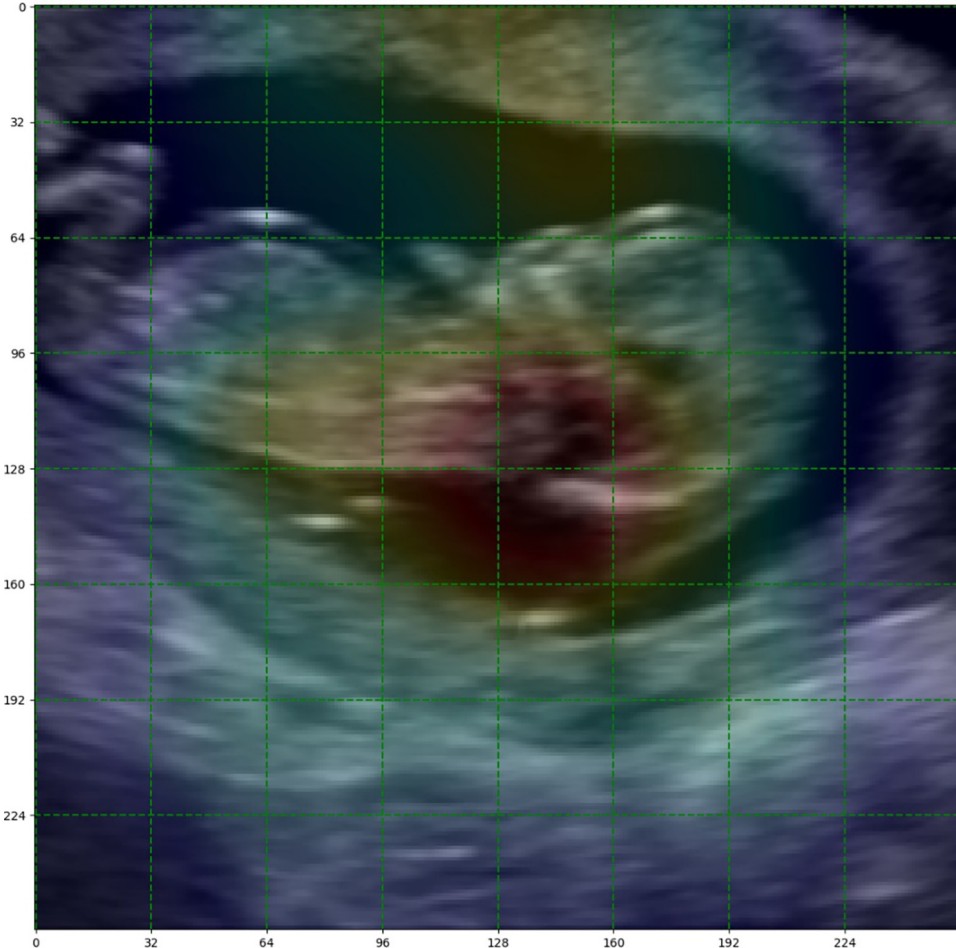

**Fig 4. Grad-CAM image of a cystic hygroma case.** The green gridlines indicate the size of the feature maps (8x8) used to generate the heat maps. The red highlights the region of the image that influenced the model's prediction the most.

## Results

The image dataset included 289 unique ultrasound images; 160 control images with normal NT measurements, and 129 cases with cystic hygroma. A total of 217 images were used for model training, and 72 images were used for model validation (Table 1).

Table 2 shows the results for all 4 cross validation folds. All 4 models performed well in the validation set. The overall mean accuracy was 93% (95% CI: 88–98%), and the area under the receiver operating characteristic curve was 0.94 (95% CI: .89–1.0) (Fig 5). The sensitivity was 92% (95% CI: 79–100%) and the specificity was 94% (95% CI: 91–96%).

Most of the Grad-CAM heat maps highlighted the fetal head and neck (Fig 6). Although some heat maps specifically highlighted the posterior cervical region, in the area used for NT measurement (Fig 7A and 7B), poor localization did occur (Fig 7C and 7D).

There were 10 false negatives and 10 false positives. The misclassified images were reviewed with a clinical expert (MW) and it was determined that misclassifications commonly happened

**Table 1. Partitioning of data across training and validation datasets[a].**

|  | Overall, n (%) | Normal NT images, n (%) | Cystic hygroma images, n (%) |
|---|---|---|---|
| **Total dataset** | 289 (100%) | 160 (100%) | 129 (100%) |
| **Training dataset** | 217 (75.1%) | 120 (75%) | 97 (75.2%) [b] |
| **Validation dataset** | 72 (24.9%) | 40 (25%) | 32 (24.8%) |

NT, nuchal translucency.

[a]Column statistics are provided.

[b]97 original images; 23 cystic hygroma images were randomly resampled to reduce imbalance between the two groups in the training dataset, to produce in a final cystic hygroma training dataset of 120 images.

when the fetus was close to the placental membrane leading the heat map to focus on another part of the brain.

## Discussion

Our findings demonstrate the feasibility of using deep-learning models to interpret fetal ultrasound images and identify cystic hygroma diagnoses with high performance in a dataset of first trimester ultrasound scans. The model achieved excellent prediction of cystic hygroma with a sensitivity of 92% (95% CI: 79–100%) and specificity of 94% (95% CI: 91–96%). This study contributes to the literature on AI and medical diagnostics, and more specifically to the use of AI in fetal ultrasonography where there are scant data.

Ultrasound is critical in the observation of fetal growth and development, however, small fetal structures, involuntary fetal movements and poor image quality make neonatal image acquisition and interpretation challenging. Our study has shown that deep-learning and Grad-CAM heat maps can correctly identify the fetal head and neck region to identify abnormalities. There have been several investigations focusing on AI-based localization of standard planes in fetal ultrasonography, suggesting that AI-models perform as well as clinicians in obtaining reasonable planes for image capture and diagnosis [23–27]. Building on this literature, cystic hygroma has not been evaluated yet it is an ideal fetal diagnostic condition to assess the accuracy of AI-based models because it is a clearly visible diagnosis to the trained expert.

More recently, others have sought to apply machine learning methods to the identification of fetal malformations, with promising results. Xie et al. developed and tested a CNN-based

**Table 2. 4-fold cross validation results[a].**

|  | FOLD NUMBER | | | | OVERALL PERFORMANCE | |
|---|---|---|---|---|---|---|
|  | Fold 0 | Fold 1 | Fold 2 | Fold 3 | Mean±SD | 95% Confidence Interval |
| **True positives, n** | 27 | 29 | 32 | 31 | 29.75±1.92 | 26.0–33.5 |
| **True negatives, n** | 38 | 38 | 37 | 37 | 37.50±0.50 | 36.5–38.5 |
| **False positives, n** | 2 | 2 | 3 | 3 | 2.50±0.50 | 1.5–3.5 |
| **False negatives, n** | 6 | 3 | 0 | 1 | 2.50±2.29 | 0.0–7.0 |
| **Accuracy** | 0.89 | 0.93 | 0.96 | 0.94 | 0.93±0.03 | 0.88–0.98 |
| **Sensitivity** | 0.82 | 0.91 | 1.00 | 0.97 | 0.92±0.07 | 0.79–1.0 |
| **Specificity** | 0.95 | 0.95 | 0.93 | 0.93 | 0.94±0.01 | 0.91–0.96 |
| **AUC** | 0.91 | 0.93 | 0.98 | 0.95 | 0.94±0.03 | 0.89–1.0 |

AUC, area under the curve; SD, standard deviation.

[a]Decision threshold in the receiver-operating characteristic (ROC) curve was set at 0.5 in the [0,1] range of predicted probability (or class membership).

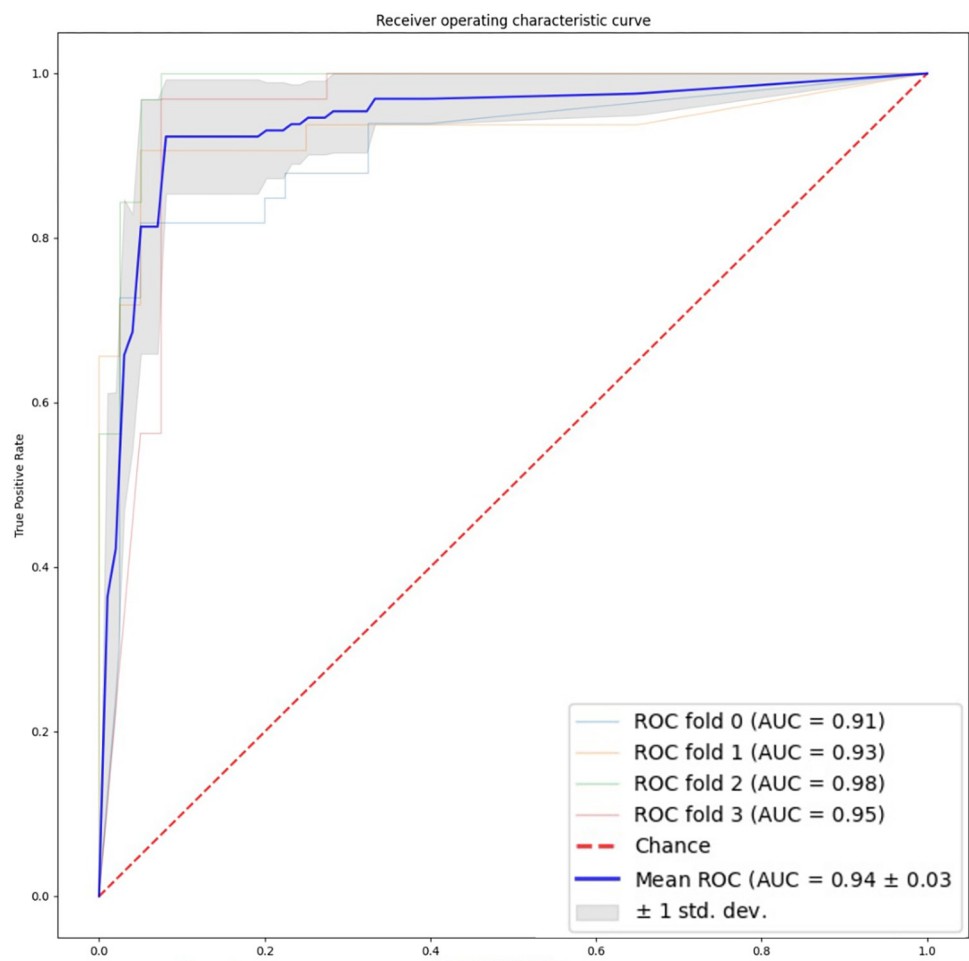

**Fig 5. Receiver operating characteristic plot summarizing performance of all four cross validation folds.**

deep-learning model in a dataset of nearly 30,000 fetal brain images, including over 14,000 images with common central nervous system abnormalities [28]. Although, the final model was able to discriminate well between normal and abnormal images (sensitivity and specificity were 96.9% and 95.9%, respectively) in a hold out test set, the models were not trained to distinguish between specific brain abnormalities and did not have information on cystic hygroma diagnoses. In a large sample of 2D ultrasounds, fetal echocardiograms and anatomy scans from second trimester pregnancies, Arnaout et al. trained CNN models to accurately localize the fetal heart and detect complex congenital heart disease [29]. Their models achieved high sensitivity (95%, 95%CI, 84–99), and specificity (96%, 95%CI, 95–97) and were validated in several independent datasets. Finally, Baumgartner et al. [25] demonstrated the potential for such systems to operate in real-time. Their findings suggest that deep-learning models could be used "live" to guide sonographers with image capture and recognition during clinical practice. The model proposed by Baumgartner et al. was trained for detection of multiple fetal standard views in freehand 2D ultrasound data. They achieved a 91% recall (i.e., sensitivity) on the profile standard views which is close to the evaluation metrics obtained in our study. Although published data on deep-learning models for the identification of specific malformations are generally lacking, our findings, combined with those of others in this space demonstrate the feasibility of deep-learning models for supporting diagnostic decision-making.

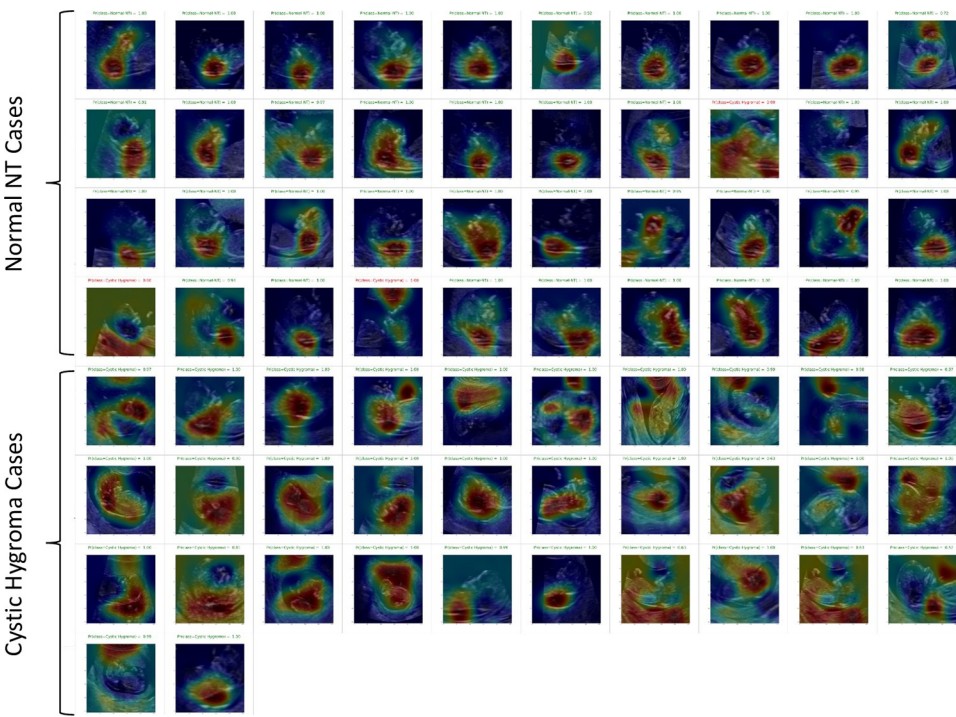

**Fig 6. Grad-CAM heat maps for the full validation set of Fold 2.** Top 4 rows are normal NT cases and bottom 4 rows are cystic hygroma cases. Red colours highlight regions of high importance and blue colours highlight regions of low or no importance. Therefore, a good model would have Grad-CAM heatmaps that highlight the head and neck area for both normal and cystic hygroma images.

Using the latest data augmentation and image dimensionality techniques, we have developed a deep-learning algorithm with very good predictive accuracy on a relatively small dataset. The strengths of this study include our use of the k-fold cross-validation experiment design. Although computationally intensive, k-fold cross-validation reduces model bias and variance, as most of the data are used for model training and validation. In addition, removal of calipers and text annotations from ultrasound images established a dataset free from the clinical bias on which to develop our models. Annotations on routinely collected ultrasound images have historically limited their utility for medical AI research [30]. Furthermore, our use of Grad-CAM heat maps enabled transparent reporting of how the deep-learning models performed on a case-by-case basis. With this approach, we were able to confirm excellent localization for most of the images used in our dataset. Class-discriminative visualization enables the user to understand where models fail (i.e., why the models predicted what they predicted) and can be used to inform downstream model enhancements. Additionally, all false negative and false positive images were reviewed and it was determined that misclassifications commonly occurred when the fetus was close to the placental membrane. Future work could collect more images where the fetus is close to membrane, up-sample the images that were error prone, and further train the model to incorporate this pattern.

Our study is not without limitations. First, as a single centre study, the sample size available for developing and validating deep-learning model was relatively small. However, use of data augmentation to increase the variability in our dataset, enrichment of cystic hygroma cases in our training set, and use of the k-fold cross validation experiment design are all well-accepted strategies to overcome the limitations of small datasets [31]. Second, although we removed all image annotations, we cannot discount the possibility that the blending and infill methods used to reconstitute the

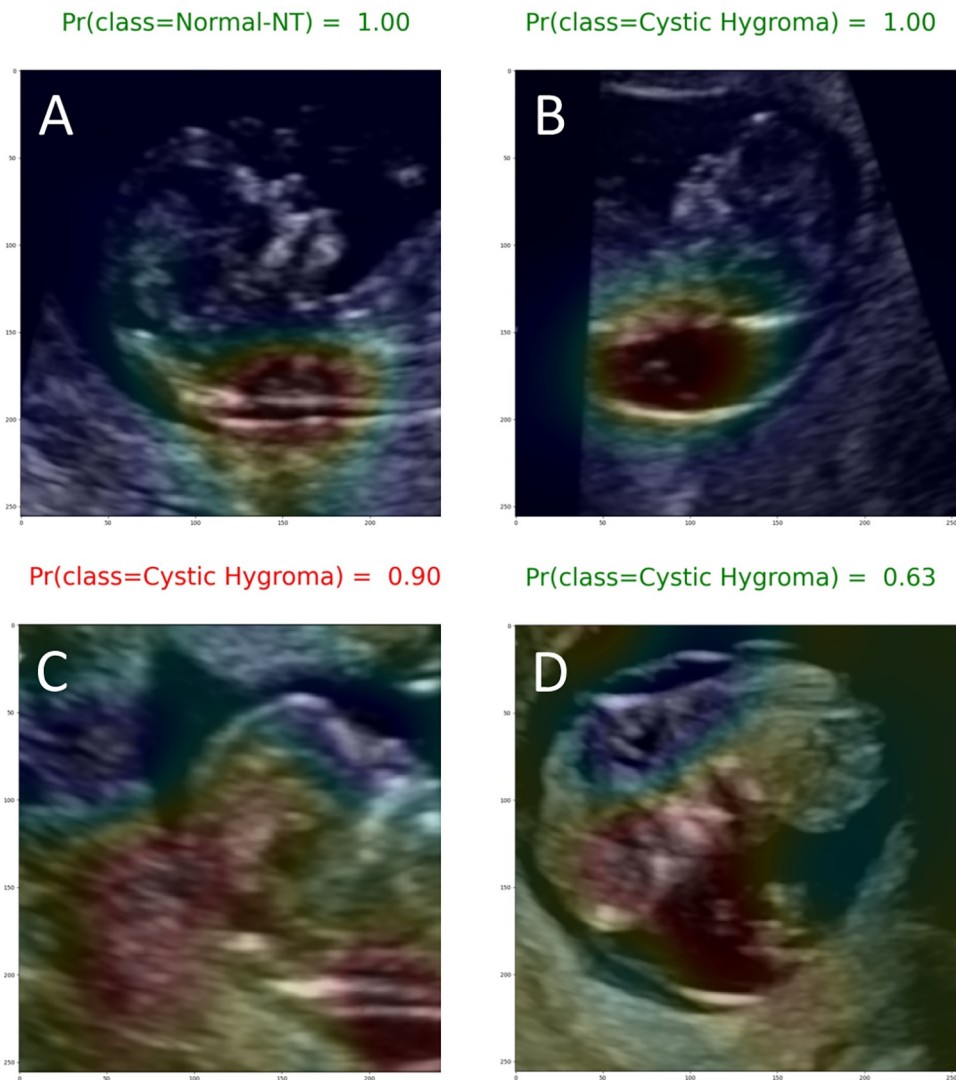

**Fig 7. Exemplary Grad-CAM heat maps.** (A) Normal NT case with good localization in which the model predicted the correct class with a high (1.00) output probability (true negative). (B) Cystic hygroma case with good localization in which the model predicted the correct class with a high (1.00) output probability (true positive). (C) Normal NT case showing poor localization in which the model predicted this class incorrectly with a 0.90 output probability (false positive). (D) Cystic hygroma case showing poor localization in which the model predicted the correct class, but with an output probability that suggests uncertainty (0.63) (true positive).

images influenced the deep-learning algorithm. However, the Grad-CAM heatmaps provide reassurance that fetal craniocervical regions were driving the deep-learning algorithm, and that the model appropriately places high importance on regions which are clinically relevant for diagnosis. Given the relatively low incidence of congenital anomalies such as cystic hygroma, a natural extension of this work will be to introduce our models to a larger, multi-centre dataset with more variability in the image parameters and greater feature variety specific to cystic hygroma.

## Conclusions

In this proof-of-concept study, we demonstrate the potential for deep-learning to support early and reliable identification of cystic hygroma from first trimester ultrasound scans. We

present a novel application of convolutional neural networks to automatically identify cases of cystic hygroma and localise the relevant fetal structures for clinical decision-making. With further development, including testing in a large multi-site dataset and external validation, our approach may be applied to a range of other fetal anomalies typically identified by ultrasonography.

## Acknowledgments

The authors would like to acknowledge that this study took place on unceded Algonquin Anishinabe territory.

## Author Contributions

**Conceptualization:** Mark C. Walker, Olivier X. Miguel, Steven Hawken.

**Data curation:** Inbal Willner, Olivier X. Miguel.

**Formal analysis:** Olivier X. Miguel, Steven Hawken.

**Funding acquisition:** Mark C. Walker.

**Investigation:** Mark C. Walker, Inbal Willner, Darine El-Chaâr, Felipe Moretti, André M. Carrington, Steven Hawken, Richard I. Aviv.

**Methodology:** Katherine A. Muldoon, André M. Carrington, Steven Hawken.

**Project administration:** Alysha L. J. Dingwall Harvey, Ruth Rennicks White.

**Resources:** Mark C. Walker.

**Supervision:** Mark C. Walker, Darine El-Chaâr, Felipe Moretti, Alysha L. J. Dingwall Harvey, Ruth Rennicks White, André M. Carrington, Steven Hawken.

**Writing – original draft:** Olivier X. Miguel, Malia S. Q. Murphy, Katherine A. Muldoon.

**Writing – review & editing:** Mark C. Walker, Inbal Willner, Olivier X. Miguel, Malia S. Q. Murphy, Darine El-Chaâr, Felipe Moretti, Alysha L. J. Dingwall Harvey, Ruth Rennicks White, Katherine A. Muldoon, André M. Carrington, Steven Hawken, Richard I. Aviv.

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
