## [Decision Letter · Decision Letter 0]

25 Mar 2022

PONE-D-21-40543Using deep-learning algorithms in fetal ultrasound analysis for diagnosis of cystic hygroma in the first trimesterPLOS ONE

Dear Dr. Walker,

Thank you for submitting your manuscript to PLOS ONE. After careful consideration, we feel that it has merit but does not fully meet PLOS ONE’s publication criteria as it currently stands. Therefore, we invite you to submit a revised version of the manuscript that addresses the points raised during the review process.

The positioning of the paper against the state of the art needs to be more thoroughly addressed. Also the experiments (setup + results) need further clarification. The choice for selecting the DenseNet169 model is not explained; at least another different architecture should be applied in comparison, in order to highlight DenseNet169 as being the better alternative. The interpretation of the visual part (human + GradCAM) is not detailed. The cases that were erroneously predicted should be discussed.

We look forward to receiving your revised manuscript.

Kind regards,

Ruxandra Stoean

Academic Editor

PLOS ONE

Journal Requirements:

4. Thank you for stating the following in the Acknowledgments Section of your manuscript: " This study was supported by a Canadian Institutes of Health Research Foundation Grant (FDN 148438). The funding agency was not involved in study design, analysis or interpretation of data. The agency was not involved writing of this manuscript or in the decision to submit the article for publication"

"This study was supported by a Canadian Institutes of Health Research Foundation Grant (FDN 148438)(MCW). The funding agency payed no role in the study design, data collection and analysis, decision to publish, or preparation of the manuscript."

Reviewers' comments:

Reviewer's Responses to Questions

**Comments to the Author**

1. Is the manuscript technically sound, and do the data support the conclusions?

Reviewer #1: Partly

Reviewer #2: No

Reviewer #3: Yes

2. Has the statistical analysis been performed appropriately and rigorously? 

Reviewer #1: No

Reviewer #2: No

Reviewer #3: Yes

3. Have the authors made all data underlying the findings in their manuscript fully available?

Reviewer #1: No

Reviewer #2: No

Reviewer #3: No

4. Is the manuscript presented in an intelligible fashion and written in standard English?

Reviewer #1: Yes

Reviewer #2: Yes

Reviewer #3: Yes

5. Review Comments to the Author

Reviewer #1: The proposed work deals with a DenseNet architecture for Cystic hygroma diagnosis in fetal ultrasound images. The paper is technically intermediate; I have the following issues that need to be resolved before acceptance.

1) The title of the paper is "Using Deep Learning Algorithms ...", but, the authors used only one DenseNet169 architecture. The name should be changed.

2) If there are other related works, then cite and describe them, what is the novelty of your work compared to such works.

3) In the Study Setup and Design section, it is mentioned that a 4-fold cross-validation was used instead of a 10-fold cross-validation. For clarity, show learning curves to better describe the behavior of the model.

4) On line 97, it written that “four different deep learning models were tested and trained…”, but only the DenseNet169 architecture was used in the work. This needs to be clarified.

5) Explain why, among all known architectures, the DenseNet169 architecture was used, give arguments for such a choice. How were the hyper-parameters chosen?

6) It is necessary to clarify the details of the architecture. So the DenseNet169 model was trained from scratch? Have the architecture weights been fine-tuned? From which layer?

7) References used in the discussion section should be cited first in Introduction. The discussion of the results should be clarified.

Reviewer #2: In the Model Training, the authors did not describe the detials of method of training the CNN model by using the utrasound image, as well as did not describe the structure of the CNN. If the authors just applied the CNN model ,which is trained by other peoples, to classify images as “normal” or those with “cystic hygroma”, I think this way is not suitable for the medical image, because the medical images are different from nature image. The author should build the network by themselves, including make the image label for the model training.

Reviewer #3: This manuscript described a study used DenseNet to classify fetal ultrasound image for diagnosis of cystic hygroma. The proposed method for the application is reasonable and the results were promising and expected. In this study, the authors also used Grad-CAM to demonstrate the explainability of deep learning model. This manuscript was well organized.

I have three concerns to this manuscript:

1. If the diagnosis of cystic hygroma is easy for human eyes, what is the effect of having such a model?

2. From the Fig. 5, the red color regions in normal and cystic hygroma seem at same location (head and neck). Thus the authors should explain how to interpret the difference between normal and cystic hygroma.

3. The number of error cases in the study is small. The authors should discuss what happened in these cases.

6. PLOS authors have the option to publish the peer review history of their article (what does this mean?). If published, this will include your full peer review and any attached files.

Reviewer #1: No

Reviewer #2: No

Reviewer #3: No

---

## [Author Response · Author response to Decision Letter 0]

8 May 2022

Response to Reviewers

General comments:

The positioning of the paper against the state of the art needs to be more thoroughly addressed. Also the experiments (setup + results) need further clarification. The choice for selecting the DenseNet169 model is not explained; at least another different architecture should be applied in comparison, in order to highlight DenseNet169 as being the better alternative. The interpretation of the visual part (human + GradCAM) is not detailed. The cases that were erroneously predicted should be discussed

• Thank you for raising these points for clarification. We have clarified that we did not conduct a comparison paper on multiple architectures. Our study was designed to assess if DenseNet169 is an appropriate architecture, rather than how it compares to other architectures. We chose DenseNet169 because it has more connections than other architectures (i.e. ResNet).

• Interpretation of the visual part (human +GradCam) is not detailed – We have expanded on this issue. All cystic hygroma images had been diagnosed in ultrasound reports and each image was abstracted and confirmed by a clinical expert independent review (IW). The following text has been added:

o ‘Cases were images if there was a final clinical diagnosis of a cystic hygroma in the ultrasound report. A set of normal ultrasound images from NT screens were used for controls and were retrieved between March 2021 and June 2021. All images were reviewed and verified by a clinical expert (IW).’(Methods, Study setting and design)

• There were 10 false negatives and 10 false positives. The misclassified images were reviewed with an independent clinical expert (MCW) and it was determined that misclassifications commonly happened when the fetus was close to the placental membrane leading the heat map to focus on another part of the anatomy. To improve this, future work could collect more images that have fetus close to membrane and upsample the images that were error prone. The following text has been added: 

o ‘There were 10 false negatives and 10 false positives. The misclassified images were reviewed with a clinical expert (MW) and it was determined that misclassifications commonly happened when the fetus was close to the placental membrane leading the heat map to focus on another part of the brain.’(Results)

• Thank you. The manuscript has been reviewed and formatted with PLOS ONE style requirements 

• Thank you for raising these points for clarification. This study was a retrospective study of ultrasound images and consent was waived. We have included the following text in the manuscript:

o ‘Patients were not contacted, and patient consent was not required to access the images. Data were de-identified and fully anonymized for the model training. This study was reviewed and approved by the Ottawa Health Sciences Network Research Ethics Board (OHSN REB #20210079).’ (Methods)

• Thank you for raising these points for clarification. This study was a retrospective study of ultrasound images and all images were fully anonymized prior to model training. We have included the following text in the manuscript:

o ‘Patients were not contacted, and patient consent was no required to access the images. Data were de-identified and fully anonymized for the model training. This study was reviewed and approved by the Ottawa Health Sciences Network Research Ethics Board (OHSN REB #20210079).’ (Methods)

• Thank you for raising these points for clarification. All information on funding has been removed from the manuscript and is only included in the online submission form.

4. Thank you for stating the following in the Acknowledgments Section of your manuscript: " This study was supported by a Canadian Institutes of Health Research Foundation Grant (FDN 148438). The funding agency was not involved in study design, analysis or interpretation of data. The agency was not involved writing of this manuscript or in the decision to submit the article for publication". We note that you have provided funding information that is not currently declared in your Funding Statement. However, funding information should not appear in the Acknowledgments section or other areas of your manuscript. We will only publish funding information present in the Funding Statement section of the online submission form. 

• Thank you for raising these points for clarification. All information on funding has been removed from the manuscript and is only included in the online submission form.

• Thank you for raising this point. The ultrasound images are not currently available to upload to PLOS ONE. All images belong to The Ottawa Hospital. A formal request can be made to the IRB, who would then consult a privacy officer at The Ottawa Hospital. Requests can be sent to the corresponding author (Dr. Mark Walker) or the Ottawa Health Sciences Network Research Ethics Board (rebadministration@ohri.ca). We have also included this information in the cover letter

• Thank you for updating the data availability statement on our behalf.

6. PLOS requires an ORCID iD for the corresponding author in Editorial Manager on papers submitted after December 6th, 2016. Please ensure that you have an ORCID iD and that it is validated in Editorial Manager. 

• Thank you for raising this point. The ORCID ID has been added for the corresponding author.

Reviewer #1: The proposed work deals with a DenseNet architecture for Cystic hygroma diagnosis in fetal ultrasound images. The paper is technically intermediate; I have the following issues that need to be resolved before acceptance.

1) The title of the paper is "Using Deep Learning Algorithms ...", but, the authors used only one DenseNet169 architecture. The name should be changed.

• Thank you for highlighting this point. We have changed the title to ‘Using deep-learning in fetal ultrasound analysis for diagnosis of cystic hygroma in the first trimester’.

2) If there are other related works, then cite and describe them, what is the novelty of your work compared to such works.

• Thank you for highlighting this point. We have expanded the discussion section to highlight other related works and emphasize the novelty of our project. Briefly, the main novelty of this study is the application of deep-learning and explainable AI techniques specifically for cystic hygroma diagnosis. Other works in the field used similar technical methods (deep-learning + GradCAM) but with different conditions (i.e. medical diagnoses not cystic hygroma). The references below have been emphasized in the manuscript:

o Baumgartner CF, Kamnitsas K, Matthew J, Fletcher TP, Smith S, Koch LM, Kainz B, Rueckert D. SonoNet: Real-Time Detection and Localisation of Fetal Standard Scan Planes in Freehand Ultrasound. IEEE Transactions on Medical Imaging. 2017;36(11):2204-2215. doi:10.1109/TMI.2017.2712367

o Xie HN, Wang N, He M, Zhang LH, Cai HM, Xian JB, Lin MF, Zheng J, Yang YZ. Using deep‐learning algorithms to classify fetal brain ultrasound images as normal or abnormal. Ultrasound in Obstetrics & Gynecology. 2020;56(4):579-587. doi:10.1002/uog.21967

o Arnaout R, Curran L, Zhao Y, Levine JC, Chinn E, Moon-Grady AJ. An ensemble of neural networks provides expert-level prenatal detection of complex congenital heart disease. Nature Medicine. 2021;27(5):882-891. doi:10.1038/s41591-021-01342-5

3) In the Study Setup and Design section, it is mentioned that a 4-fold cross-validation was used instead of a 10-fold cross-validation. For clarity, show learning curves to better describe the behavior of the model.

• Thank you for raising these points for clarification. We conducted a 4-fold cross-validation. The 4-fold cross-validation (4CV) design was chosen instead of the more commonly used 10-fold cross-validation (10CV) design to optimize the performance of the deep-learning models within our small dataset. With 4CV, each prediction error affects the accuracy of the model by 1.4%, and the sensitivity by 3.1%. Had a 10CV approach been used, each prediction error would have affected model accuracy by 3.4% and the sensitivity by 7.8%.

4) On line 97, it written that “four different deep learning models were tested and trained…”, but only the DenseNet169 architecture was used in the work. This needs to be clarified.

• Thank you for raising this point. We conducted a 4-fold cross-validation where every fold in the cross-validation experiment trains one DenseNet169 architecture. Therefore, each fold produces 1 trained model yielding a total of 4 different DenseNet169 models (i.e. same architecture, trained on 4 folds, producing 4 different models).

• The following text has been added:

o ‘A 4-fold cross-validation (4CV) design was used, whereby the same deep-learning architecture was tested and trained four different times using randomly partitioned versions (folds) of the image dataset. For each fold, 75% of the dataset was used for model training, and 25% was used for model validation. The 4CV design was chosen instead of the more commonly used 10-fold cross-validation (10CV) design to optimize the performance of the deep-learning models within our small dataset. With 4CV, each prediction error affects the accuracy of the model by 1.4%, and the sensitivity by 3.1%. Had a 10CV approach been used, each prediction error would have affected model accuracy by 3.4% and the sensitivity by 7.8%.’ (Methods, Study setting and design)

5) Explain why, among all known architectures, the DenseNet169 architecture was used, give arguments for such a choice. How were the hyper-parameters chosen?

• Thank you for raising these points for clarification. We have expanded on this in the methods section. The following text has been added:

o ‘DenseNet169 was chosen because of its advantages over the ResNet architecture as shown in [12]. On some image classification benchmarks DenseNet demonstrated better performance and efficiency than the ResNet model.’ (Methods, Neural Network Model)

o ‘Default hyperparameters were chosen and no hyperparameters tuning was conducted. Batch size was chosen to maximize graphic processing units (GPU) memory usage. Learning rate (lr) scheduler parameters were chosen to cover lr range of 0.01 to 0.0005. Large lr (0.01) are good for initialization of the model but will not converge as well as smaller lr (0.0005). Therefore, a decreasing lr is used to get the best of both large and small lrs.’ (Methods, Model Training)

6) It is necessary to clarify the details of the architecture. So, the DenseNet169 model was trained from scratch? Have the architecture weights been fine-tuned? From which layer?

• Thank you for raising these points for clarification. Yes, the DenseNet169 was trained from scratch and did not use pretrained weights. Pretrained DenseNet169 architecture was not available for grayscale images and potentially would not have worked appropriately with medical ultrasound images. We trained the architecture using our dataset of ultrasound images from The Ottawa Hospital with random weights initialization.

• The following text has been added to the manuscript:

o ‘Pretrained DenseNet169 architecture was not available for grayscale images and potentially would not have worked appropriately with medical ultrasound images. We trained the architecture using our dataset of ultrasound images from The Ottawa Hospital with random weights initialization.’(Methods, Model Training)

7) References used in the discussion section should be cited first in Introduction. The discussion of the results should be clarified.

• Thank you for highlighting this point. We have adjusted to introduction to include the references. 

Reviewer #2: In the Model Training, the authors did not describe the details of method of training the CNN model by using the ultrasound image, as well as did not describe the structure of the CNN. If the authors just applied the CNN model ,which is trained by other peoples, to classify images as “normal” or those with “cystic hygroma”, I think this way is not suitable for the medical image, because the medical images are different from nature image. The author should build the network by themselves, including make the image label for the model training.

• Thank you for raising these points for consideration. This is an important issue that was raised with Reviewer 1 as well. We have expanded this section in detail and including more information on the method for training the CNN model and choice of architecture. Please see the comments to Reviewer 1 in Questions 3,4,5 and 6

• We have referenced the full description of the architecture of the DenseNet for readers who want to know more. See comments to reviewer 1 question 5 for explanation of our decision to use DenseNet.

• We used off-the-shelves DenseNet169 model trained from scratch. No pretrained weights and no fine-tuning was done. In other words, we built the dataset (made the image label) and trained the model ourselves (trained from scratch). We did not use a model trained by other people (ex: pretrained on ImageNet dataset) for two main reasons: 1) Models pretrained with ultrasound images were not readily available; 2) Medical images such as ultrasound are significantly different from natural images (eg. RGB vs grayscale, ultrasound artefacts different than natural image artefacts etc.) therefore models pretrained with natural images could be unsuitable. 

Reviewer #3: This manuscript described a study used DenseNet to classify fetal ultrasound image for diagnosis of cystic hygroma. The proposed method for the application is reasonable and the results were promising and expected. In this study, the authors also used Grad-CAM to demonstrate the explainability of deep learning model. This manuscript was well organized. I have three concerns to this manuscript:

1. If the diagnosis of cystic hygroma is easy for human eyes, what is the effect of having such a model?

• Thank you for raising these points for clarification. Through fetal ultrasound, cystic hygroma is an easily identifiable congenital anomaly for trained experts. We began our pilot study with a fetal anomaly with a clearly visible diagnosis to assess the feasibility of using deep learning with ultrasounds before moving on to more challenging diagnoses. The following text has been included in the manuscript:

o ‘There have been several investigations focusing on AI-based localization of standard planes in fetal ultrasonography, suggesting that AI-models perform as well as clinicians in obtaining reasonable planes for image capture and diagnosis,[21–25]. Building on this literature, cystic hygroma has not been evaluated yet it is an ideal fetal diagnostic condition to assess the accuracy of AI-based models because it is a clearly visible diagnosis to the trained expert.’(Discussion)

2. From the Fig. 5, the red color regions in normal and cystic hygroma seem at same location (head and neck). Thus, the authors should explain how to interpret the difference between normal and cystic hygroma.

• Thank you for raising these points for clarification. The head and neck area are the correct area to distinguish cystic hygroma from normal. In figure 5, the GradCAM confirmed that the convolutional neural network was giving attention to the correct anatomical location for cystic hygroma ascertainment. Therefore, a good model would have GradCAM heatmaps that highlight the head and neck area for both normal and cystic hygroma images since a normal image is the absence of the disease.

• The following text has been added:

o ‘Fig 5. Grad-CAM heat maps for the full validation set of Fold 2. Top 4 rows are normal NT cases and bottom 4 rows are cystic hygroma cases. Red colours highlight regions of high importance and blue colours highlight regions of low or no importance. Therefore, a good model would have GradCAM heatmaps that highlight the head and neck area for both normal and cystic hygroma images.’ (Results, Figure 5).

• The following text has been added to clarify how to interpret the difference between normal and cystic hygroma

o ‘Cystic hygroma is characterized by increased nuchal translucency (i.e. fluid under the skin behind the fetal neck) with visible septations. Diagnosis is straightforward and possible through routine first- or second-trimester ultrasonography. Affected pregnancies require extensive antenatal and postpartum management which may include early cytogenetic testing for suspected aneuploidy, comprehensive assessment of fetal anatomy, and postpartum surgical intervention.’ (Introduction)

3. The number of error cases in the study is small. The authors should discuss what happened in these cases.

• Thank you for raising this point. In this pilot study there were 10 false negative images and 10 false positives. Each image was reviewed by a clinical expert (MW) and it was noted that misclassifications often happened when the fetus was close to the placental membrane leading the heat map to focus away from the head and neck region and on to another part of the body To improve this, future work could collect more images where the fetus is close to membrane, up-sample the images that were error prone, and further train the model to incorporate this pattern. 

• The following text has been added to the manuscript:

o ‘There were 10 false negatives and 10 false positives. The misclassified images were reviewed with a clinical expert (MCW) and it was determined that misclassifications commonly happened when the fetus was close to the placental membrane leading the heat map to focus on another part of the body.’(Results, Figure 6).

o ‘Class-discriminative visualization enables the user to understand where models fail (i.e., why the models predicted what they predicted) and can be used to inform downstream model enhancements. Additionally, all false negative and false positive images were reviewed and it was determined that misclassifications commonly occurred when the fetus was close to the placental membrane. Future work could collect more images where the fetus is close to membrane, up-sample the images that were error prone, and further train the model to incorporate this pattern.’ (Discussion)

---

## [Decision Letter · Decision Letter 1]

19 May 2022

Using deep-learning in fetal ultrasound analysis for diagnosis of cystic hygroma in the first trimester

PONE-D-21-40543R1

Dear Dr. Walker,

We’re pleased to inform you that your manuscript has been judged scientifically suitable for publication and will be formally accepted for publication once it meets all outstanding technical requirements.

Kind regards,

Ruxandra Stoean

Academic Editor

PLOS ONE

Additional Editor Comments (optional):

Reviewers' comments:

Reviewer's Responses to Questions

**Comments to the Author**

1. If the authors have adequately addressed your comments raised in a previous round of review and you feel that this manuscript is now acceptable for publication, you may indicate that here to bypass the “Comments to the Author” section, enter your conflict of interest statement in the “Confidential to Editor” section, and submit your "Accept" recommendation.

Reviewer #1: All comments have been addressed

Reviewer #3: All comments have been addressed

2. Is the manuscript technically sound, and do the data support the conclusions?

Reviewer #1: Yes

Reviewer #3: Yes

3. Has the statistical analysis been performed appropriately and rigorously? 

Reviewer #1: Yes

Reviewer #3: Yes

4. Have the authors made all data underlying the findings in their manuscript fully available?

Reviewer #1: Yes

Reviewer #3: No

5. Is the manuscript presented in an intelligible fashion and written in standard English?

Reviewer #1: Yes

Reviewer #3: Yes

6. Review Comments to the Author

Reviewer #1: (No Response)

Reviewer #3: The authors had addressed my concerns and the manuscript had been revised according to review commnets. I have no further comments.

7. PLOS authors have the option to publish the peer review history of their article (what does this mean?). If published, this will include your full peer review and any attached files.

Reviewer #1: No

Reviewer #3: No

---

## [Editor Report · Acceptance letter]

26 May 2022

PONE-D-21-40543R1 

Using deep-learning in fetal ultrasound analysis for diagnosis of cystic hygroma in the first trimester 

Dear Dr. Walker:

I'm pleased to inform you that your manuscript has been deemed suitable for publication in PLOS ONE. Congratulations! Your manuscript is now with our production department. 

Kind regards, 

on behalf of

Dr. Ruxandra Stoean 

Academic Editor

PLOS ONE